# Resurrection of 2′-5′-oligoadenylate synthetase 1 (OAS1) from the ancestor of modern horseshoe bats blocks SARS-CoV-2 replication

Spyros Lytras[1]*, Arthur Wickenhagen[1], Elena Sugrue[1], Douglas G. Stewart[1], Simon Swingler[1], Anna Sims[1], Hollie Jackson Ireland[1], Emma L. Davies[1], Eliza M. Ludlam[1], Zhuonan Li[1], Joseph Hughes[1], Sam J. Wilson[1,2]*

1 MRC–University of Glasgow Centre for Virus Research, University of Glasgow, Glasgow, United Kingdom, 2 Cambridge Institute of Therapeutic Immunology & Infectious Disease (CITIID), Jeffrey Cheah Biomedical Centre, Department of Medicine, University of Cambridge, Cambridge, United Kingdom

☉ These authors contributed equally to this work.
* spyros.lytras@glasgow.ac.uk (SL); sjw58@cam.ac.uk (SJW)

**Data Availability Statement:** Raw experimental data and images are within the paper's Supporting Information (S1 Data and S2 Data). All data and

## Abstract

The prenylated form of the human 2′-5′-oligoadenylate synthetase 1 (OAS1) protein has been shown to potently inhibit the replication of Severe Acute Respiratory Syndrome Coronavirus 2 (SARS-CoV-2), the virus responsible for the Coronavirus Disease 2019 (COVID-19) pandemic. However, the OAS1 orthologue in the horseshoe bats (superfamily Rhinolophoidea), the reservoir host of SARS-related coronaviruses (SARSr-CoVs), has lost the prenylation signal required for this antiviral activity. Herein, we used an ancestral state reconstruction approach to predict and reconstitute in vitro, the most likely OAS1 protein sequence expressed by the Rhinolophoidea common ancestor prior to its prenylation loss (RhinoCA OAS1). We exogenously expressed the ancient bat protein in vitro to show that, unlike its non-prenylated horseshoe bat descendants, RhinoCA OAS1 successfully blocks SARS-CoV-2 replication. Using protein structure predictions in combination with evolutionary hypothesis testing methods, we highlight sites under unique diversifying selection specific to OAS1's evolution in the Rhinolophoidea. These sites are located near the RNA-binding region and the C-terminal end of the protein where the prenylation signal would have been. Our results confirm that OAS1 prenylation loss at the base of the Rhinolophoidea clade ablated the ability of OAS1 to restrict SARSr-CoV replication and that subsequent evolution of the gene in these bats likely favoured an alternative function. These findings can advance our understanding of the tightly linked association between SARSr-CoVs and horseshoe bats.

code relating to computational analysis are provided in the following GitHub online repository: https://github.com/spyros-lytras/ancient_bat_OAS1 and the Zenodo repository with DOI: 10.5281/zenodo.10022254.

**Funding:** This study was supported by Medical Research Council (https://www.ukri.org/councils/mrc) awards MR/P022642/1 (SJW), MR/V01157X/1 (SJW) and MC_UU_12014/12 (JH). SS holds a Daphne Jackson Fellowship funded by Medical Research Scotland. The funders had no role in study design, data collection and analysis, decision to publish, or preparation of the manuscript.

**Competing interests:** The authors have declared that no competing interests exist.

**Abbreviations:** ASR, ancestral sequence reconstruction; COVID-19, Coronavirus Disease 2019; CPE, cytopathic effect; DMEM, Dulbecco's Modified Eagle's Medium; DMV, double-membrane vesicle; FCS, fetal calf serum; IFN, interferon; ISG, interferon-stimulated gene; OAS1, 2′-5′-oligoadenylate synthetase 1; SARS-CoV-2, Severe Acute Respiratory Syndrome Coronavirus 2; SARSr-CoV, SARS-related coronavirus.

## Introduction

To date, there are at least 10 coronaviruses known to have transmitted from an animal reservoir to humans, 3 of which have a putative rodent origin (HKU1) potentially through a cattle intermediate (OC43 and HECV-4408), 5 likely originating in bats (229E, NL63, MERS-CoV, SARS-CoV, and SARS-CoV-2), one of a canine-feline origin (CCoV-HuPn-2018), and one of porcine origin (Hu-PDCoV) [1–5]. These range from old spillovers, now endemic in humans, to recent epidemic and pandemic viruses, to one-off animal-to-human transmission chains. Severe Acute Respiratory Syndrome Coronavirus 2 (SARS-CoV-2), has had the greatest documented impact on global health, being responsible for the Coronavirus Disease 2019 (COVID-19) pandemic, and its origins can be traced to horseshoe bats of the genus *Rhinolophus* [2,6]. Multiple barriers prevent viruses from infecting new species and these barriers can also hamper the ability of a virus to thrive in an unfamiliar host. Alternatively, changes in the host genes responsible for these barriers may enable some viruses to diversify in these host lineages. Functional engagement of dependency factors (such as viral receptors) is required for successful spillover, but viruses must also navigate a myriad of host-specific immune defences to sustain infection and transmission in a new host. The prenylated form of the human 2′-5′-oligoadenylate synthetase 1 (OAS1, isoform p46) protein has been shown to have potent antiviral activity against SARS-CoV-2 and its expression correlates with less severe disease in humans [7–11]. The OASs are interferon-stimulated genes (ISGs) and their expression levels are commonly increased during interferon (IFN)-mediated antiviral responses. Most OASs sense double-stranded viral RNA, and this frequently activates the synthesis of 2′-5′-linked oligoadenylates (2-5A); 2-5A induces the dimerization of inactive RNase L, which upon activation mediates the indiscriminate cleavage of viral and host RNAs, leading to inhibition of viral replication [12,13]. The mammalian OAS family includes 3 catalytically active members (OAS1, OAS2, and OAS3) [14] that seem to be under co-adaptive evolutionary pressure with its interacting proteins (RNase L and STING) in both primates and Chiroptera [15].

For the OAS1 protein to recognise the virus and initiate its inhibition, it needs to be in contact with the viral RNA. Coronaviruses hijack the endoplasmic reticulum of infected cells, creating double-membrane vesicles (DMVs) in which virus genome replication takes place [16,17]. Similar replication mechanisms are deployed by the majority of positive-sense single-stranded RNA (+ssRNA) viruses [18]. This intracellular compartmentalisation of the viral genetic material may be a form of immune evasion and suggests that host antiviral proteins also need to be localised in (or near) the DMVs to target the virus. The p46 isoform of human OAS1 contains a prenylation signal at its C-terminal end that targets the protein to membranes in close proximity to where SARS-CoV-2 replicates thereby enabling antiviral activity. In contrast, the non-prenylated p42 isoform does not restrict SARS-CoV-2 replication [9,10]. Similar posttranslational modification signals seem to be required by other RNA-binding ISG proteins for sensing RNA viruses replicating in cytoplasmic compartments and inhibiting their replication [19]. We have previously shown that the CAAX box prenylation motif required for anti-SARS-CoV-2 activity has been ablated by an ancestral LTR insertion shared by all sequenced horseshoe bat genomes (superfamily Rhinolophoidea) [10].

The presumed absence of OAS1-dependent anti-CoV activity in the Rhinolophoidea could remove the selective pressure on the coronaviruses that typically infect them to evade or antagonise this defence. Notably, lineage A *Betacoronaviruses* (such as OC43 and MHV) and MERS coronaviruses (lineage C) have independently acquired phosphodiesterase (PDE)-encoding genes (NS2 and NS4b, respectively) that antagonise the OAS-RNase L antiviral pathway [20–23]. Indeed, the majority of PDE-encoding betacoronaviruses infect non-Rhinolophoidea bats, while we have shown that no coronavirus sampled in a Rhinolophoidea bat so far encodes any

PDE-like proteins [10]. If the LTR insertion deleting the OAS1 prenylation signal in the Rhinolophoidea common ancestor removed its (and its descendants') ability to restrict coronavirus replication through the OAS1-RNAse L pathway, then this event could partially explain why SARS-related, non-PDE-encoding coronaviruses seem to ancestrally circulate in bat species of the Rhinolophoidea superfamily.

In this study, we explore how the OAS1 protein lost its presumed inhibitory activity against SARS-related coronaviruses at the base of the Rhinolophoidea superfamily and provide novel insights about the evolution of this protein in horseshoe bats. To do this, we phylogenetically reconstructed the ancestral OAS1 gene sequence likely present in the Rhinolophoidea common ancestor at the time of the LTR insertion. We show that by adding the prenylation signal at the end of the ancestrally reconstructed protein, we recover antiviral activity against SARS-CoV-2. This is not the case for the extant *Rhinolophus ferrumequinum* OAS1 protein with an appended prenylation signal. We have used an array of selection analyses to identify sites under diversifying selection unique to the post-LTR, Rhinolophoidea OAS1 evolution. Mapping the sites under selection to the predicted protein structure of a horseshoe bat OAS1 indicates potential changes in RNA binding specificity and overall change of protein function in this superfamily.

## Results

### The Rhinolophoidea common ancestor OAS1 protein

Our current understanding of the deeper phylogenetic relation of bats (order Chiroptera) splits them into 2 major suborders: the Yinpterochiroptera (including superfamilies Rhinolophoidea and Pteropodoidea) and the Yangochiroptera (including superfamilies Noctilionoidea and Verspertilionoidea) [24]. The LTR insertion deleting the OAS1 prenylation signal is shared between all Rhinolophoidea members with available genome sequences. This means that the deletion and putative loss of OAS1 anti-SARS-CoV-2 function took place after the split between the Rhinolophoidea and Pteropodoidea superfamilies and prior to the diversification of the extant Rhinolophoidea species (Fig 1A). In addition to OAS1 of all other bat taxa having retained the prenylation signal, SARS-CoV-2 restriction has been confirmed in vitro using OAS1 from members of the Pteropodoidea (*Pteropus alecto*) and the Yangochiroptera (*Pipistrellus kuhlii*) as well as humans, camels, cows, and mice [10]. Hence, the antiviral sensing of CoV dsRNA mediated by prenylated OAS1 is likely the ancestral phenotype, also shared by the deep ancestor of all Rhinolophoidea (Fig 1A), prior to the LTR insertion.

We retrieved a set of 18 Chiroptera OAS1 protein sequences—available from NCBI GenBank or reconstructed from whole genome assemblies in this study—and used a phylogenetic approach, informed by the Chiroptera species tree, to predict the sequence of the aforementioned Rhinolophoidea pre-insertion ancestor OAS1 protein. The method used here provides a posterior probability for each site, indicative of the confidence on the reconstructed state. The majority of sites were confidently predicted with a posterior probability above 0.95. As expected, sites with lower posteriors corresponded to variable positions on the Chiroptera alignment (posteriors inversely correlating with position entropy—S1 Fig).

Although most ancestral sequence reconstruction (ASR) methods are useful for predicting the state of single informative sites of internal nodes, indel variation in alignments can prove problematic in sequence reconstruction [25]. By examining the OAS1 protein alignment, we identified a short region with distinct indel variation between bat taxa, corresponding to *R. ferrumequinum* OAS1 amino acid positions 159 to 163 (Fig 1A). Members of the Pteropodoidea and the Noctilionoidea have the longest variable region sharing clear homology, despite the 2 superfamilies not being monophyletic. This indicates that the longest genotype is likely the

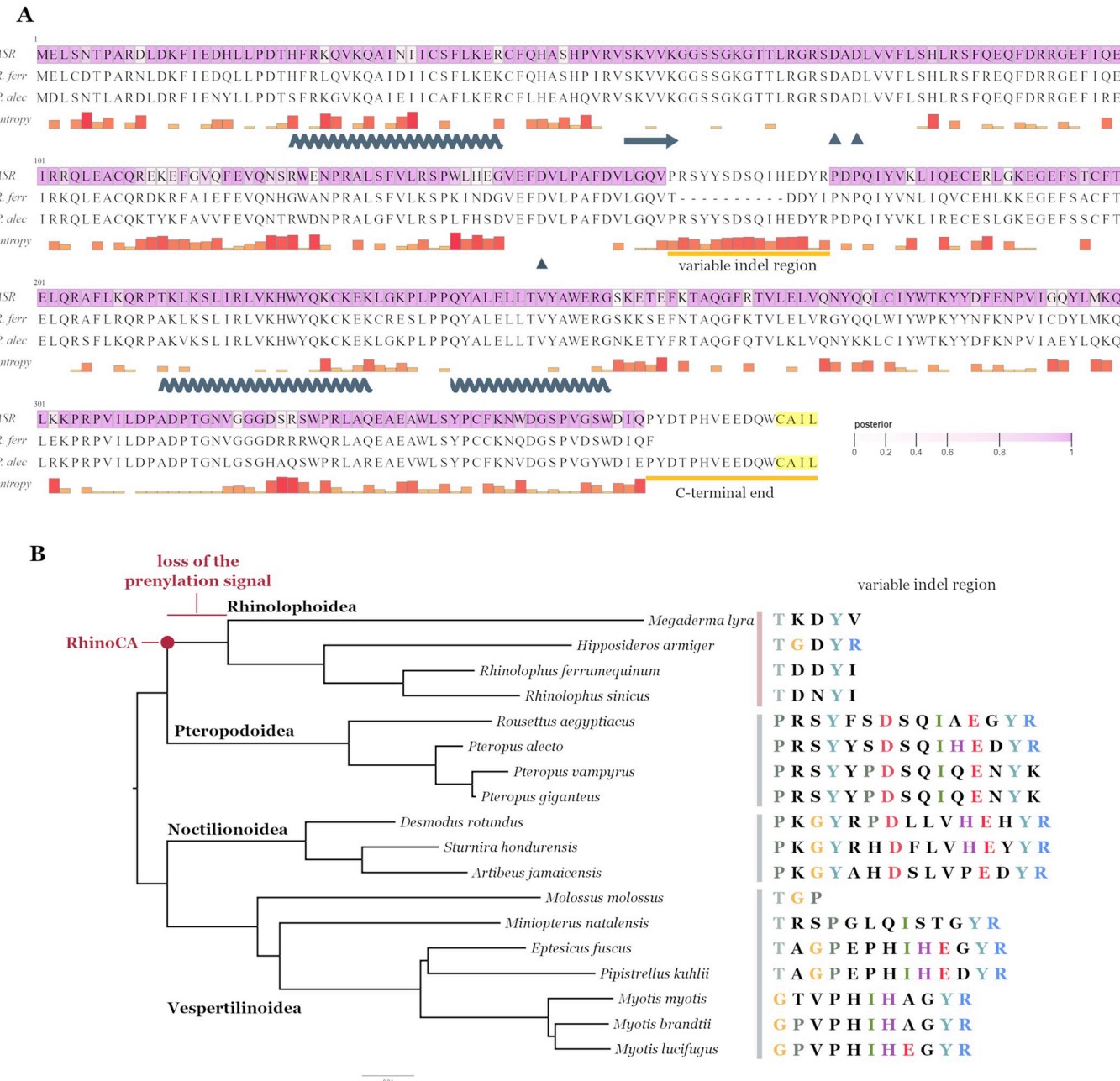

**Fig 1. Ancestral state reconstruction of the RhinoCA OAS1 sequence.** (A) Protein alignment of the RhinoCA (ASR), *R. ferrumequinum* (R. ferr), and *P. alecto* (P. alec) OAS1 sequences. RhinoCA sites are coloured by each predicted state's posterior probability. The bars on the bottom row indicate the Shannon's entropy of each site in the alignment of all 18 Chiroptera OAS1 proteins. Secondary structure alpha helices (zigzag) and beta sheets (arrows) involved in the protein/RNA interface and the active site triad residues D74/D75/D147 are annotated underneath the entropy row (as described for the human OAS1 protein in Donovan and colleagues [26]). The CAAX box signal at the C-terminal end of the sequence is highlighted in yellow. (B) Maximum likelihood phylogeny of the Chiroptera OAS1 proteins informed by their species tree topology. The ancestrally reconstructed (RhinoCA) node, the branch where the prenylation signal was deleted and the clades of each superfamily are annotated on the tree. The variable indel region sequence of OAS1 is shown on the right of each tip. Residues are coloured according to potential homology between the proteins. ASR, ancestral sequence reconstruction; OAS1, 2′-5′-oligoadenylate synthetase 1.

ancestral state of all bats, with the Vespertilionoidea having undergone short deletions in the region (with the exception of the *Molossus molossus* OAS1) while Rhinolophoidea have lost most of this region (Fig 1B). To account for this region in the original sequence reconstruction,

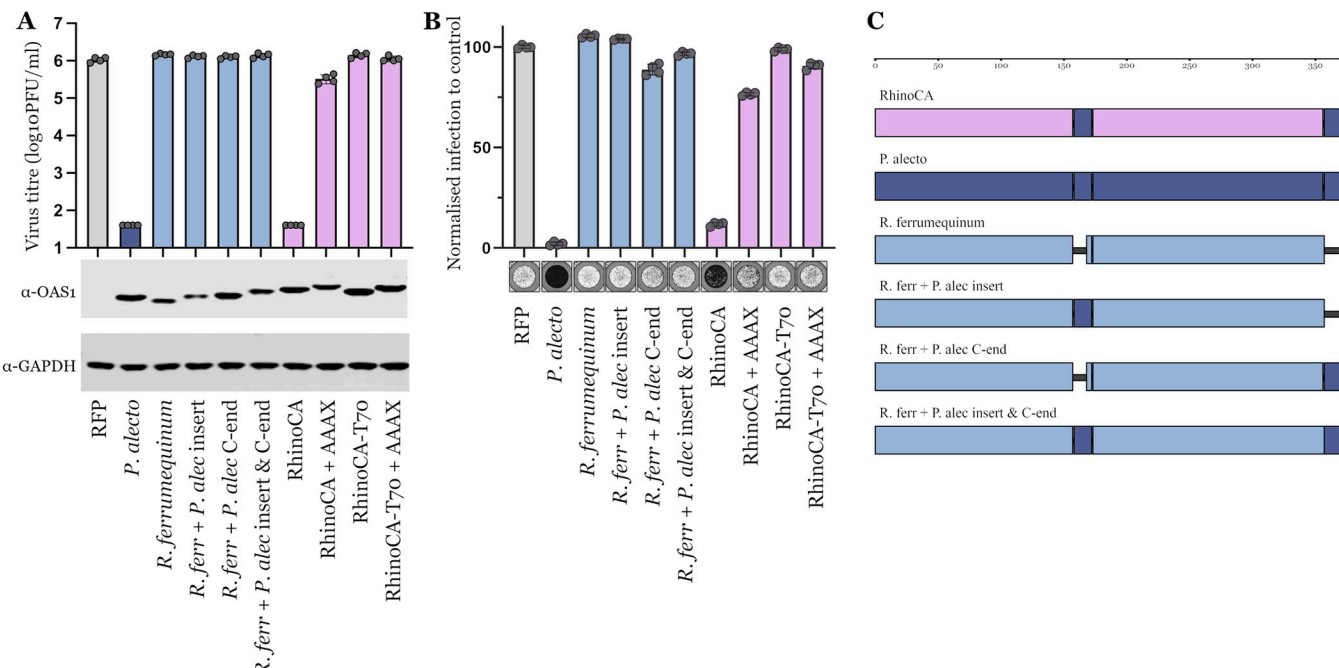

**Fig 2. RhinoCA restricts SARS-CoV-2 replication.** (A) Infectious titers of SARS-CoV-2 (PFU/ml) determined on AAT cells (A549-ACE2-TMPRSS2) modified to expressing bat OAS1 proteins (*P. alecto*, *R. ferrumequinum*), their specified derivatives, and the ancestrally reconstructed RhinoCA and RhinoCA-T70 sequences. Controls for the ASR genes include identical sequences with ablated prenylation motifs (CAAX terminal end changed to AAAX). OAS1 expression was monitored in parallel by western blot (S2 Data). (B) SARS-CoV-2 infection on AAT cells expressing exogenous OAS1 constructs as in panel A (based on well clearance caused by CPEs of virus replication). Infection normalized to RFP control and a typical picture of virus-induced CPE is shown below each graph. Values for all replicates of both assays are presented in S1 Data. (C) Illustration of bat OAS1 recombinant constructs tested in panels A and B. ASR, ancestral sequence reconstruction; CPE, cytopathic effect; OAS1, 2′-5′-oligoadenylate synthetase 1; SARS-CoV-2, Severe Acute Respiratory Syndrome Coronavirus 2.

we removed the site-by-site predicted segment and replaced it with the corresponding sequence of the *P. alecto* OAS1. This choice was based on the following: (i) the longest region genotype likely being ancestral to all bats; (ii) Pteropodoidea being the clade most closely related to the Rhinolophoidea; and (iii) having confirmed that the *P. alecto* OAS1 restricts SARS-CoV-2 in vitro (suggesting this region was unlikely to negatively impact reconstructed OAS1 antiviral activity).

Similarly, the C-terminal end of the ancestral OAS1 sequence could not be reconstructed since most of the region is deleted in the Rhinolophoidea species and there is high indel variation between the rest of the bat OAS1s (online supplementary). To match the variable indel region insertion, the C-terminal end of the *P. alecto* OAS1 (known to initiate a block to SARS-CoV-2) was appended to the reconstructed sequence to complete the ancestral sequence. We refer to this Rhinolophoidea common ancestor OAS1 protein as "RhinoCA" OAS1 (Fig 2C). Although it is impossible to know the true OAS1 protein sequence expressed by the Rhinolophoidea common ancestor, RhinoCA is the best prediction of the extinct protein sequence, given the available data. Since some sites of the RhinoCA ASR were not confidently predicted (low state posterior), we also implemented an alternative strategy where all sites with a posterior below 0.7 were replaced with the corresponding residues of the *P. alecto* OAS1 sequence. This alternative ASR is referred to as RhinoCA-T70. The RhinoCA and RhinoCA-T70 ancient OAS1 sequences differed from the extant *R. ferrumequinum* OAS1 protein at 57 and 61 protein sites, respectively (excluding gaps), representing approximately 60 million years of evolution [10].

## Restored anti-SARS-CoV-2 activity in the ancestral OAS1 protein

After reconstructing the RhinoCA and RhinoCA-T70 OAS1 sequences, we tested whether exogenous expression of these proteins could initiate a block to SARS-CoV-2 replication. We show that expression of the ancestral RhinoCA OAS1 in A549-ACE2-TMPRSS2 cells [27] potently inhibited SARS-CoV-2, resulting in more than a 4-log reduction in virus titre (Fig 2A) and a 90% reduction of cytopathic effect (CPE)-induced well clearance [27] (Fig 2B) compared to cells expressing an RFP control. This antiviral phenotype is equivalent to that of the *P. alecto* OAS1 (Fig 2A and 2B) as well as of *P. kuhlii* and human p46 OAS1 proteins [10]. On the contrary, overexpressing the *R. ferrumequinum* OAS1 sequence had no effect on virus replication [10]. Similarly, ablating the prenylation signal of the RhinoCA OAS1 protein with a single amino acid change (CAAX to AAAX) effectively ablated antiviral activity and rescued virus replication, confirming that prenylation is essential for the antiviral activity (Fig 2A and 2B).

We questioned whether recapitulation of the antiviral phenotype was simply due to the addition of the *P. alecto* prenylated C-terminal end or the insertion at the gene's internal variable region, rather than the ancestral site reconstruction. To test this, we created recombinant OAS1 sequences of the *R. ferrumequinum* protein with (i) only the prenylated *P. alecto* C-terminal end; (ii) only the *P. alecto* variable region insertion; or (iii) both the insertion and C-terminal end (Fig 2C). None of the recombinant genotypes restored virus inhibition in the *R. ferrumequinum* protein (Fig 2A and 2B), consistent with the hypothesis that the ancestral loss of the prenylation site was followed by divergence of function over the past 60 million years of Rhinolophoidea diversification. It should be noted that when the CAAX box sequence is appended to the human p42 OAS1 isoform there is partial rescue of the SARS-CoV-2 restriction phenotype and prenylated p42 can inhibit SARS-CoV-2 plaque formation by more than 100-fold [10]. In contrast, none of the *R. ferrumequinum* recombinant proteins initiated a block to SARS-CoV-2 replication, supporting the notion that OAS1 function has changed in the Rhinolophoidea, in a way that is not functionally analogous to the human p42 protein.

Interestingly, RhinoCA-T70 did not restrict SARS-CoV-2 replication (Fig 2A and 2B). This further demonstrates that the anti-CoV ability of bat OAS1 proteins is not solely dependent on the presence of a prenylation signal. Instead, amino acid variation also defines the presence or absence of anti-SARS-CoV-2 activity. The RhinoCA and RhinoCA-T70 differ in 16 sites that do not cluster in any specific part of the protein (S1 Table and Fig 3A). The ambiguously predicted sites of RhinoCA-T70 were replaced by *P. alecto* OAS1 residues, hence, these residues might not follow the true evolutionary tree expectation. Since the *P. alecto* OAS1 does restrict SARS-CoV-2, its corresponding residues only disrupt antiviral function when placed in the ancestrally predicted backbone, suggestive of epistatic interactions between multiple sites controlling function. To investigate the residue changes separating the RhinoCA and Rhino-CA-T70 sequences in silico, we first predicted the tertiary structure of the RhinoCA protein based on its sequence and inferred the structure's electrostatic potential map. Out of the 16 candidate residues, site 34 seems to directly interact with the dsRNA molecule, being part of a long alpha helix next to the binding site (Figs 1A and 3B). Residue changes on the nearby site 28 of the human OAS1 protein (site 27 on the RhinoCA protein) are known to be detrimental for RNA-binding activity [26]. RhinoCA has an asparagine (N) on site 34, which has an uncharged chain, while RhinoCA-T70 has a negatively charged glutamic acid (E) instead (Fig 3B). Given that RNA is negatively charged, changing site 34 may disrupt RNA binding under this conformation, making it the most likely single change culprit for RhinoCA-T70's lack of antiviral function. We proceeded to test this hypothesis in vitro by infecting cell lines exogenously expressing the 2 proteins with swapped site 34 residues (RhinoCA 34E and Rhi-noCA-T70 34N; Fig 3E). Consistent with our hypothesis, only changing the site 34 residue is

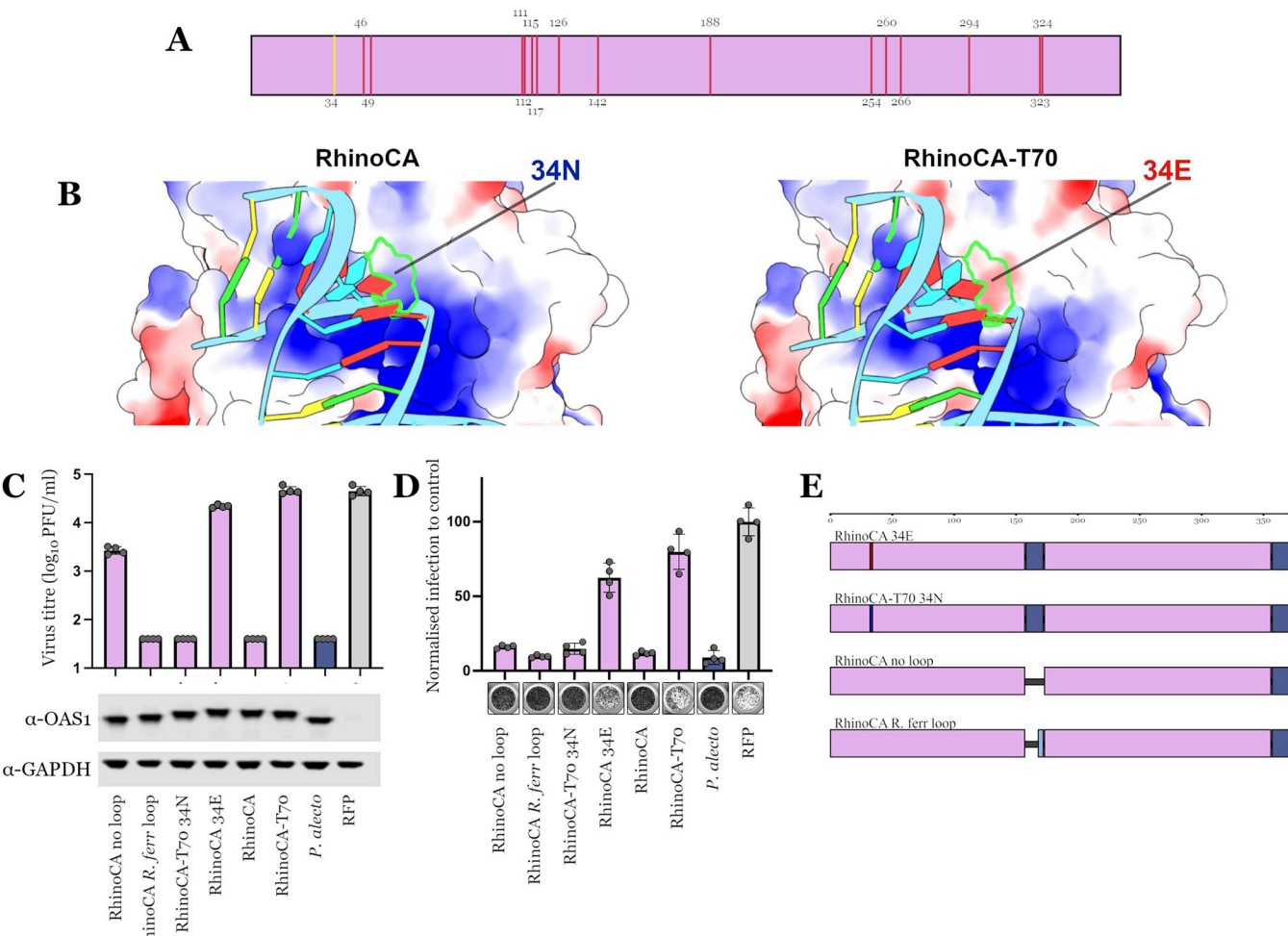

**Fig 3. Amino acid differences between the RhinoCA and RhinoCA-T70 sequence reconstructions and the relevance of the variable indel region on anti-CoV function.** (A) Schematic of amino acid differences between RhinoCA and RhinoCA-T70 on the secondary sequence structure. (B) Electrostatic potential prediction calculated with ChimeraX [28] on the RhinoCA structural protein model with an asparagine residue (left) and a glutamic acid residue (right) on site 34. (C) Infectious titers of SARS-CoV-2 (PFU/ml) determined on AAT cells (A549-ACE2-TMPRSS2) modified to expressing the *P. alecto* OAS1 protein, the ancestrally reconstructed RhinoCA and RhinoCA-T70 sequences and their specified derivatives. OAS1 expression was monitored in parallel by sestern blot (S2 Data). (D) SARS-CoV-2 infection on AAT cells expressing exogenous OAS1 constructs as in panel C (based on well clearance caused by CPEs of virus replication). Infection normalised to RFP control and a typical picture of virus-induced CPE is shown below each graph. Values for all replicates of both assays are presented in S1 Data. (E) Illustration of modified RhinoCA and RhinoCA-T70 OAS1 constructs tested in panels C and D. CPE, cytopathic effect; OAS1, 2′-5′-oligoadenylate synthetase 1; SARS-CoV-2, Severe Acute Respiratory Syndrome Coronavirus 2.

sufficient for reversing the proteins' SARS-CoV-2 inhibition phenotype (Fig 3C and 3D). The 34N RhinoCA-T70 protein shows the same magnitude of virus inhibition as the original RhinoCA protein, fully restoring the phenotype. However, the 34E RhinoCA protein still shows a small inhibitory effect compared to the RFP control and original RhinoCA-T70 in both the plaque and well-clearance assays (Fig 3C and 3D). This could suggest that RhinoCA residues at sites other than 34 may additionally contribute to the antiviral phenotype in combination with 34N.

The RhinoCA OAS1 protein contains the longer, 15 amino acid, variable indel region present in the *P. alecto* OAS1 (Fig 1) that does not restore SARS-CoV-2 inhibition when inserted to the *R. ferrumequinum* OAS1 (Fig 2). However, it is not known whether this distinct genetic feature of bat OAS1s plays any role in the proteins' antiviral activity at all. Hence, we

constructed 2 additional versions of the RhinoCA protein with the variable indel region removed completely ("RhinoCA no loop") or replaced with the shorter, 5 amino acid, genotype present in the *R. ferrumequinum* ("RhinoCA R. ferr loop") (Fig 3E). Surprisingly, replacing the *P. alecto* insertion with the *R. ferrumequinum* one on RhinoCA OAS1 (RhinoCA R. ferr loop) has no effect on the protein's antiviral activity against SARS-CoV-2 (Fig 3C and 3D). Removing the variable indel region all together (RhinoCA no loop) shows somewhat less robust inhibition of SARS-CoV-2 replication when exogenously expressed compared to RhinoCA and RhinoCA R. ferr loop. There is approximately 5% more well clearance (Fig 3D) and a more than 1-log reduction in viral titre compared to the near 4-log to control produced by the other 2 conditions (Fig 3C). Notably, all known bat OAS1s have a loop containing at least 3 amino acids in this region (Fig 1B) suggesting that even a minimal loop is beneficial for the function of OAS1 in bats. Overall, the variable indel region does not seem to directly affect the proteins' anti-CoV activity.

## Unique evolutionary signatures following prenylation loss

Following the loss of OAS1 prenylation at the basal branch of the Rhinolophoidea, we hypothesised that this OAS1 lineage might have taken an evolutionary path distinct to other bat OAS1s, such as lack of conservation of residues needed for the anti-CoV function or selection for a different function entirely. To assess differences in selective pressures of individual branches across the entire Chiroptera OAS1 phylogeny, we first used the aBSREL method [29]. Three branches in the tree show evidence of significant episodic diversifying selection: (i) the ancestral branch leading to the Yangochiroptera clade ($p = 0.024$) which—considering that both *P. alecto* and *P. kuhlii* OAS1 proteins restrict SARS-CoV-2 replication—is unlikely to have selective changes related to gain or loss of antiviral function; (ii) the terminal branch leading to *M. molossus* ($p = 0.0099$), associated with changes unique to this distant species; and (iii) the branch leading to the *Rhinolophus* clade (consisting of *R. ferrumequinum* and *R. sinicus*; $p = 0.018$). Diversification on the latter branch could be associated with divergence of protein function in this non-prenylated group. Still, no episodic selection was detected on the branch where prenylation loss took place, suggesting no major advantageous substitutions happened immediately after the loss of membrane targeting.

Since the *R. ferrumequinum* OAS1 antiviral function cannot be restored simply by appending a prenylation signal at its C-terminal end, subsequent changes on the genome likely occurred that removed this function. Similarly, RhinoCA-T70 only has 16 amino acids different to RhinoCA, some of which disrupt anti-SARS-CoV-2 function. Hence, the branches of the Rhinolophoidea clade might have undergone relaxation of potential purifying selection acting on sites required for antiviral activity in all other bat clades. The RELAX method [30] showed no evidence of selection relaxation specific to this clade (K = 0.92, $p = 0.38$, LR = 0.77) compared to the rest of the tree. Consistent with this finding, the contrast-FEL method [31] found no sites in the alignment to be evolving under a unique selective environment specific to the Rhinolophoidea clade (q value threshold of 0.2). Relaxation or change of selective pressures on this clade could have indicated a lack of significant function of the Rhinolophoidea OAS1s (or progressive pseudogenisation), however, that does not seem to be the case. Rather, the nature of selection on the Rhinolophoidea OAS1 genes has not changed substantially following the putative loss of anti-CoV function.

We then sought to understand if the sites under selection are different between the Rhinolophoidea and the other Chiroptera clades. Only testing Rhinolophoidea branches reveals 31 sites under purifying selection using the FEL method [32] and 25 sites under diversifying selection using the MEME method [33] (14 of which are also picked up by FEL; $p$ value threshold of 0.1) (S1 Table). Testing the remaining branches shows a total of 99 sites under purifying

selection (FEL) and 32 sites under diversifying selection (detected with MEME, 8 of which are also picked up by FEL; $p$ value threshold of 0.1). It is notable that although about the same number of positively selected sites is picked up in both sets of branches, only about a third has signal of purifying selection in the Rhinolophoidea branches compared to the rest of the tree. This is not a direct comparison because of differences in the number of branches tested and the amount of diversity between the 2 sets, but it could indicate that site-specific purifying selection is weaker in the Rhinolophoidea clade, hence less likely to be detected.

Comparing the identified sites between the 2 sets revealed 15 sites under diversifying selection unique to the Rhinolophoidea clade (S1 Table). These did not seem to cluster in any obvious way on the secondary structure of the protein. To examine potential clustering on the tertiary protein structure, we used AlphaFold [34] to predict structural models of the *R. ferrumequinum*, *P. alecto*, and RhinoCA OAS1 sequences. When superimposed with the human OAS1 protein structure (pdb: 4IG8), there are very few differences between the 4 structures. The 2 key distinctions of the *P. alecto* and RhinoCA OAS1s are also obvious in the sequence alignment (Fig 1A), namely: (i) the variable indel region; and (ii) the prenylated C-terminal end. The former insertion creates an unresolved loop structure located near where the dsRNA molecule binds to the protein (Fig 4A). We showed that the RhinoCA protein with different

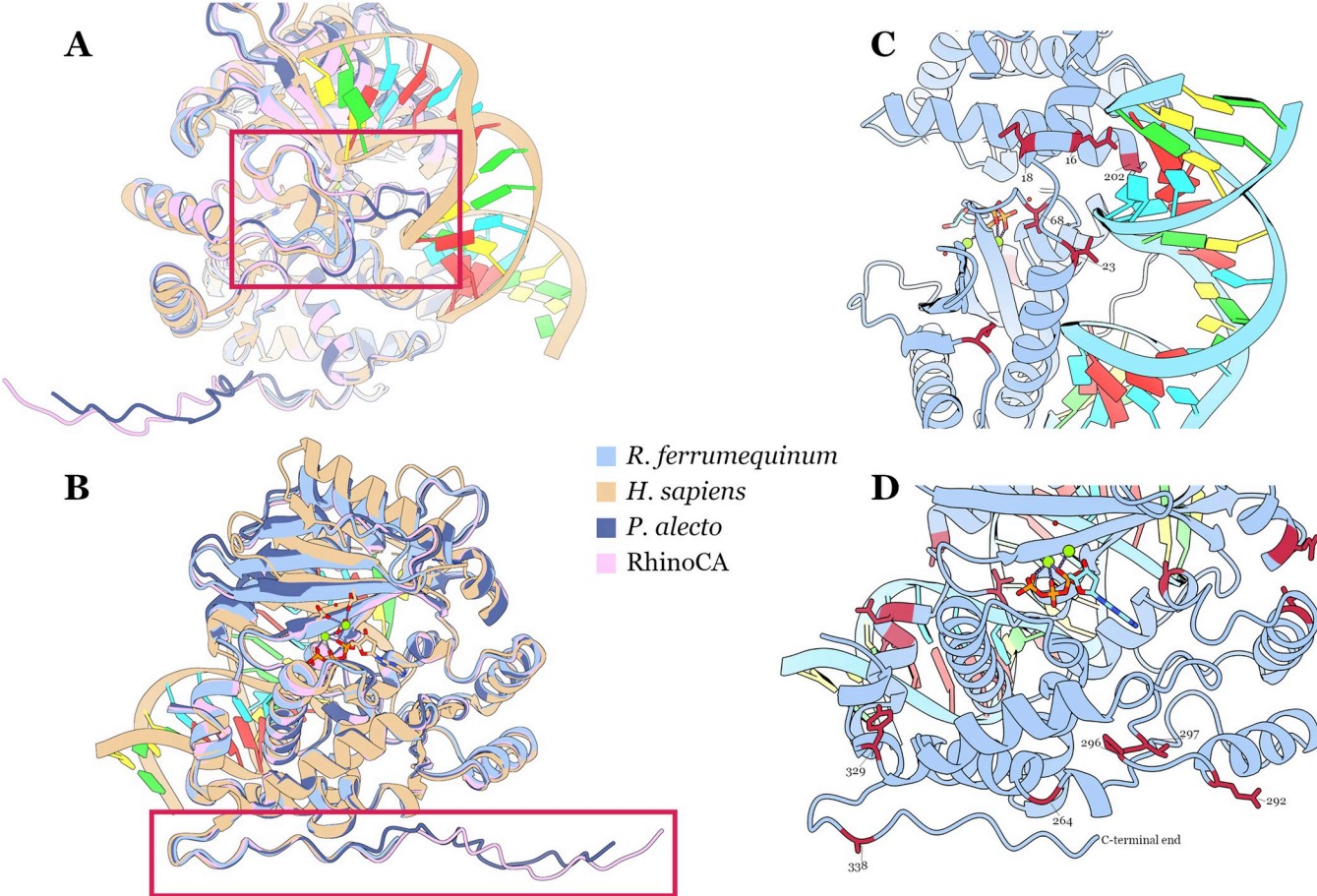

**Fig 4. Structural comparison and sites under selection unique to the Rhinolophoidea clade.** Structural models of the *R. ferrumequinum*, *P. alecto*, and RhinoCA protein sequences superimposed onto the human OAS1 structure, highlighting the variable indel region loop (A) and the prenylated C-terminal end (B). *R. ferrumequinum* OAS1 structure bound to a dsRNA helix highlighting sites under Rhinolophoidea-specific diversifying selection in red near the RNA molecule (C) and near the C-terminal end (D). OAS1, 2′-5′-oligoadenylate synthetase 1.

loop lengths maintains its anti-CoV activity (Fig 3C–3E), suggesting that the loop is unlikely to disrupt RNA binding, but might modulate binding sensitivity or stability. The latter insertion is also unresolved on the structure, but found on one end of the protein, away from the RNA-binding surface and seems to be easily accessible by enzymes for acquiring posttranslational modifications (Fig 4B). Finally, we mapped the 15 sites under positive selection unique to the Rhinolophoidea clade onto the tertiary structure of the *R. ferrumequinum* OAS1. Five of these sites (16, 18, 23, 68, and 202) potentially directly interact with the dsRNA helix (Fig 4C), suggesting that RNA-binding specificity could be under diversifying selection unique to the Rhinolophoidea. Another 6 of the sites under selection (264, 292, 296, 297, 329, and 338) cluster on the C-terminal end of the protein (Fig 4D). These all seem to face outwards of the core of the protein, exactly where the CAAX box prenylation signal would have been. We speculate that deletion of the CAAX end could have resulted in selective changes in sites located structurally near this part of the protein. No apparent function could be speculated for the remaining 4 positively selected sites (88, 116, 175, and 187).

## Discussion

The prenylated form of OAS1 is an important defence against SARS-CoV-2 in humans—the virus's novel host—and this observation is partly explained by this modification appearing to be absent in horseshoe bats—the reservoir hosts of SARS-related coronaviruses (SARSr-CoVs). Here, we reconstruct the likely sequence of the ancient OAS1 protein found in the Rhinolophoidea common ancestor, before its prenylation signal was lost. The in vitro expression of the ancient Rhinolophoidea OAS1 protein in human cells potently inhibits SARS-CoV-2 replication. This anti-CoV function cannot be restored simply by appending the prenylation signal to an extant Rhinolophoidea species' OAS1 protein, suggesting that our ASR based on the bat OAS1 phylogeny, in addition to prenylation, is responsible for restoring this function. To our knowledge, this is one of very few examples where antiviral function, lost millions of years ago, is empirically restored by reconstructing the extinct form of a gene [35].

Only one of the 2 versions of the ancestrally reconstructed OAS1 protein was capable of restricting SARS-CoV-2 replication (RhinoCA; Fig 2). However, we show that antiviral activity can be restored in the second version (RhinoCA-T70) by a single residue change—E34N—at the protein's site 34 (Fig 3). This site is located near the dsRNA-binding site (Fig 3B), and changes in nearby sites have consistently been shown to disrupt binding for the human OAS1 protein [26]. Despite the presence of 34E seemingly removing the anti-CoV activity from the ancestrally reconstructed Rhinolophoidea common ancestor OAS1, the *P. alecto* OAS1 protein contains the 34E residue and still restricts SARS-CoV-2 replication (Figs 1A, 2A and 2B). This means that the effect of amino acid substitutions to the anti-CoV activity of bat OAS1 proteins is context-dependent, suggesting strong epistatic interactions across different sites.

When first examining the alignment of all bat OAS1 proteins we observed notable, clade-specific genetic diversity in the proteins' variable indel region, hinting to potential functional importance of this part of the protein (Fig 1). However, we show that this genetic variation seems to be unrelated to the anti-CoV function, since switching or removing the variable indel region on the RhinoCA OAS1 protein has minimal to no effect on SARS-CoV-2 restriction (Fig 3C and 3D). The start of the bat variable indel region corresponds to the start of exon 3 of the human OAS1 gene. Recently, Banday and colleagues showed that another SNP in exon 3 of the human OAS1 gene associated with increased hospitalisation of COVID-19 patients produces isoforms with a shortened exon 3 start [36]. This splicing variation seems to decrease OAS1 expression through nonsense-mediated decay of shorter isoforms, likely explaining its association with more severe disease. If the variable indel region is also near a splicing site in

the bat genes, then the length variation we observe across the bats could simply represent modulation of the dominant isoform in each species, most or all producing both long and short isoforms that have not yet been identified. On the same grounds, the indel variation could affect OAS1 expression in each bat species. Our assays exogenously express only the coding sequence of the bat OAS1 proteins; hence, we are unable to determine a potential effect of the variable indel region to protein expression. At least 4 different human OAS1 isoforms are known (p42, p46, p48, and p52), suggesting that there could be unexplored isoform diversity of the bat orthologues.

Showing that OAS1 anti-CoV activity is restored at the base of the Rhinolophoidea superfamily clade supports loss of this function being due to the ancestral LTR insertion and can provide new insights on the arms race evolution between SARSr-CoVs and these bats. At least 2 distinct betacoronavirus lineages have independently acquired phosphodiesterase-encoding genes that counteract OAS1-dependent antiviral activity. Both viral lineages are thought to have ancestrally infected species expressing prenylated OAS1 proteins previously shown to restrict SARS-CoV-2 [10]: rodents or cattle for *Betacoronaviruses* in lineage A [1] and bats of the Vespertilionidae family for MERS coronaviruses in lineage C [37–39]. This suggests that PDE gene acquisition was likely selected for in their distant reservoir hosts. Having lost their OAS1 defence against coronaviruses, the early Rhinolophoidea species would have been an easily accessible niche for non-PDE-expressing CoVs, such as the SARSr-CoVs, to establish as their long-term hosts. Thus, OAS1 prenylation loss due to a stochastic LTR insertion about 60 million years ago could be one of the key reasons why SARSr-CoVs circulate in present day horseshoe bats. Previous research has demonstrated how unique evolution of other immune genes in bats has likely led to enhanced "innate immune tolerance" for these animals [40]. This could also be the result of OAS1's evolution in horseshoe bat, explaining the large diversity of SARSr-CoVs that they carry [41].

Despite the ancestral loss of anti-CoV activity, we show that selective pressures have not substantially relaxed on the Rhinolophoidea OAS1 clade. This indicates that the gene is not pseudogenising and probably has biological function that remains conserved within the superfamily. The sites under Rhinolophoidea-specific diversifying selection clustering near the RNA-binding surface and C-terminal region (Fig 4C and 4D) considered alongside the *Rhinolophus* branch selection signal, suggest that the horseshoe bat OAS1 has developed a novel function. This could be restricting viruses (where posttranslational modification of OAS1 for membrane localisation is not required) or could be a function unrelated to innate immunity. Considering the lack of knowledge of other potential isoforms produced by the bat OAS1 genes, the shift in evolutionary signatures might not be indicative of true novel function, rather changing the evolutionary focus on an existing function performed by a different isoform. The OASs are ancient proteins with extensive retention of duplications in their evolutionary histories [14] and homology dating back to the animal-insect split [42,43], so although most research has focused on their immune properties, they could be involved in other cellular functions requiring RNA sensing. Lastly, very few Rhinolophoidea bat genomes have been sequenced so far. Sequencing the OAS1 locus of more species or even acquiring population-level resolution of allele frequencies for these bats would largely enhance our understanding of this functional change in the Rhinolophoidea OAS1.

## Materials and methods

### Retrieval of bat OAS1 proteins

We used protein BLAST [44] with the *R. ferrumequinum* OAS1 protein (XP_032953023.1) as the query sequence. The search was restricted to the Chiroptera order and after manual

examination of the pairwise alignments, the OAS1 protein sequences of 16 bat species (including *R. ferrumequinum*) were retrieved.

From the Rhinolophoidea superfamily, only *R. ferrumequinum* and *Hipposideros armiger* have annotated OAS1 protein sequences available in NCBI Genbank. To increase the phylogenetic resolution of this clade, we retrieved the contigs from the *Megaderma lyra* and *R. sinicus* genomic assemblies (PVJL010007185.1, NW_017739019.1) that are syntenic to the *R. ferrumequinum* OAS1 locus (previously identified using DIGS [10,45]) and used AUGUSTUS [46] to predict the respective OAS1 coding sequences (human version with default transition matrix). Sequence predictions were aligned to the *R. ferrumequinum* OAS1 sequence using Mafft v7.453 [47] and one sequence was selected for each species, based on highest transcript similarity to the *R. ferrumequinum* OAS1 protein (XP_032953023.1).

### Ancestral sequence reconstruction

The resulting 18 bat OAS1 protein sequences were aligned with Mafft (v7.453;—genafpair option) [47] and, in order to avoid low-information sites in the alignment biasing the phylogenetic reconstruction, N- and C-terminal ends not shared by the majority of sequences were trimmed off.

Iqtree (version 1.6.1, [48]) was used for the ancestral sequence reconstruction (-*asr*) of the final protein alignment under a LG+I+G4 model (selected by ModelFinder [49]). The reconstructed phylogeny's topology was informed by the species tree of the corresponding bat species retrieved from TimeTree [50] (online supplementary), using iqtree's -*te* option.

The iqtree output was used to reconstruct the OAS1 sequence preceding the LTR insertion in the Rhinolophoidea common ancestor, i.e., the node of the phylogeny connecting the Rhinolophoidea and the Pteropodoidea superfamilies. The RhinoCA sequence was reconstructed using the residue with the highest posterior probability for each site. A second version of the sequence, RhinoCA-T70, was reconstructed by replacing all sites where no residue state had a posterior probability above 0.7 with the corresponding *P. alecto* residue. Since gaps in the alignment provide no information for the site-by-site ancestral reconstruction, the variable indel region corresponding to *R. ferrumequinum* OAS1 positions 159 to 163 was replaced with the *P. alecto* insertion in this region (*P. alecto* OAS1 positions 159–173 –PRSYYSDSQIHE-DYR) for both RhinoCA and RhinoCA-T70. Similarly, the *P. alecto* C-terminal end was appended at the C-terminal end of both reconstructed sequences (*P. alecto* OAS1 positions 357–372 –PYDTPHVEEDQWCAIL). Positions in the alignment where residues (rather than gaps) were present in only one out of the 18 bat OAS1 sequences were removed from the reconstructions.

The entropy value for each site in the Chiroptera OAS1 protein alignment shown in Fig 1A was calculated using Shannon's entropy formula with a natural log as implemented in Bioedit [51] ($H(l) = -Sf(a,l)\ln(f(a,l))$; $f(a,l)$ being the frequency of amino acid $a$ at position $l$).

### Virus infections and titrations

A549-ACE2-TMPRSS2 ("AAT") cells (described before in Wickenhagen and colleagues [10] and Rihn and colleagues [27]) were maintained in Dulbecco's Modified Eagle's Medium (DMEM) supplemented with 9% fetal calf serum (FCS) and 10 μg/ml gentamicin. The SARS-CoV-2 isolate CVR-GLA-1 was used for all SARS-CoV-2 infections under appropriate biosafety conditions and has been described previously [27].

Overexpression of genes corresponding to the cDNA of open reading frames for: *P. alecto* OAS1 (NP_001277091.1), *R. ferrumequinum* OAS1 (XP_032953023.1), and the ancestrally reconstructed constructs shown in Fig 2C and 3E (online supplementary) were synthesised as

ene blocks with flanking SfiI sites (IDT DNA) and subcloned into the lentiviral vector pLV-EF1a-IRES-Puro-SfiI-TagRFP [10]. Successful expression of the gene products in AAT cells was confirmed by western blot analysis. Briefly, cells were seeded at $10^6$ cells/well in six-well plates the day before harvest. Cells were washed once with PBS, harvested in SDS sample buffer [12.5% glycerol, 175 mM Tris-HCl (pH 8.5), 2.5% SDS, 70 mM 2-mercaptoethanol, and 0.5% bromophenol blue] and then heated for 10 min at 70˚C and sonicated. After protein separation on NuPage 4% to 12% Bis-Tris polyacrylamide gels and transfer onto nitrocellulose membranes, proteins were detected using OAS1 (rabbit polyclonal 14955-1-AP, Proteintech) or GAPDH (mouse monoclonal 60004-1-Ig, Proteintech) antibodies. Goat anti-rabbit IgG (Thermo Fisher Scientific, 35568) and goat anti-mouse IgG (Thermo Fisher Scientific, SA5-10176) fluorescently labelled secondary antibodies were used for detection on a LiCor Odyssey scanner.

Infection assays with SARS-CoV-2 (plaque assay and CPE-induced well-clearance assays) have been described before [10,27]. For plaque assays, 12-well plates were seeded with $3 \times 10^5$ cells/well of AAT derivative cells overnight. The next day, cells were inoculated with 10-fold logarithmic dilutions of virus stock and absorbed for 1 h at 37˚C. Cells were subsequently overlaid with 0.6% Avicel in MEM and incubated for 72 h. Followed by fixation in 8% formaldehyde and stained with a Coomassie blue solution for plaque visualisation. Well-clearance assays were seeded in 96-well plates at $1.25 \times 10^4$ cells/well and infected the following day with titrated 3-fold dilutions. After 72 h, cells were fixed in 8% formaldehyde and cell monolayers were stained with Coomassie blue. The assay quantifies transmitted light (Celigo, Nexcelom) that penetrates stained cell monolayers with CPE cleared wells transmitting more light than intact monolayers of protected or uninfected cells.

## Selection analysis

The trimmed amino acid alignment of the 18 bat OAS1 proteins was converted to its corresponding coding sequence alignment using pal2nal [52]. To exclude potentially non-homologous sites before performing selection analysis, the variable indel region (highlighted in Fig 1A) was removed from the alignment. The final codon alignment contained 351 out of the 366 codon sites in the original alignment. The phylogeny was reconstructed again using the gap-free codon alignment in the same way described above (-asr -te) under a GTR+I+F+G4 substitution model using iqtree [48]. The resulting phylogeny and alignment were used for performing a number of selection detection methods of the Hyphy package (v2.5.33) [53]. RELAX [30] was performed to detect potential signals of selection relaxation specific to all branches of the Rhinolophoidea clade and branch leading up to it. aBSREL [29] was used to detect branch-specific episodic selection across all branches of the tree. To examine site-specific selection, methods FEL and MEME [32,33] were performed, each with 100 permutations of parametric bootstrapping, separately for the branches of the Rhinolophoidea clade and branch leading up to it and all other branches of the tree. Contrast-FEL [31] was performed to detect sites under different selective pressures in the Rhinolophoidea using the aforementioned branches as test and reference, respectively.

## Protein structure predictions

ColabFold (https://colab.research.google.com/github/sokrypton/ColabFold/blob/main/AlphaFold2.ipynb)[54], implementing mmseq2 [55], and AlphaFold2 [34], was used to predict the tertiary structure of the *R. ferrumequinum*, *P. alecto*, and RhinoCA OAS1 proteins. ColabFold was performed under default parameters and the best ranked prediction was selected for each protein. The structures were visualised and superimposed onto the RNA-bound human OAS1 crystal structure (pdb: 4IG8) using ChimeraX (version 1.4) [28].

## Supporting information

**S1 Fig. Relation between site entropy and ASR posterior estimate.** Dot plot showing Chiroptera OAS1 alignment entropy values and RhinoCA ancestral state reconstruction posterior values for each site in the alignment. Linear regression with confidence interval shading calculated with Observable HQ plots is presented. Sites with posterior values of 0 corresponding to multiple gaps have been removed from the plot.
(TIF)

**S1 Table. Site-specific selection analysis summary.** FEL and MEME analysis site-specific selection results for all alignment sites testing only on Rhinolophoidea branches in the tree and testing on all non-Rhinolophoidea branches (Chiroptera). Ancestrally predicted residues for sequences RhinoCA and RhinoCA-T70 are also presented for each site.
(CSV)

**S1 Data. Raw experimental data.** Raw data for plaque assays and CPE assays presented in Figs 2A, 2B, 3C and 3D.
(XLSX)

**S2 Data. Raw images.** Raw images for western blots supporting the experimental assays, corresponding to Figs 2A and 3C.
(PDF)

## Acknowledgments

We are grateful to Jennifer Havens for helpful discussions on the RELAX approach.

## Author Contributions

**Conceptualization:** Spyros Lytras, Arthur Wickenhagen, Elena Sugrue, Joseph Hughes, Sam J. Wilson.

**Data curation:** Spyros Lytras, Arthur Wickenhagen, Simon Swingler, Anna Sims.

**Formal analysis:** Spyros Lytras, Arthur Wickenhagen, Elena Sugrue, Douglas G. Stewart, Simon Swingler, Anna Sims.

**Funding acquisition:** Sam J. Wilson.

**Investigation:** Spyros Lytras, Arthur Wickenhagen, Elena Sugrue, Douglas G. Stewart, Simon Swingler, Anna Sims, Hollie Jackson Ireland, Emma L. Davies, Eliza M. Ludlam, Zhuonan Li.

**Methodology:** Spyros Lytras, Arthur Wickenhagen, Elena Sugrue, Douglas G. Stewart, Simon Swingler, Anna Sims.

**Software:** Spyros Lytras.

**Supervision:** Joseph Hughes, Sam J. Wilson.

**Validation:** Spyros Lytras, Arthur Wickenhagen, Elena Sugrue, Douglas G. Stewart, Simon Swingler, Anna Sims, Hollie Jackson Ireland, Emma L. Davies, Eliza M. Ludlam, Zhuonan Li.

**Visualization:** Spyros Lytras, Arthur Wickenhagen, Douglas G. Stewart.

**Writing – original draft:** Spyros Lytras, Arthur Wickenhagen, Elena Sugrue.

**Writing – review & editing:** Spyros Lytras, Arthur Wickenhagen, Elena Sugrue, Douglas G. Stewart, Simon Swingler, Anna Sims, Hollie Jackson Ireland, Emma L. Davies, Eliza M. Ludlam, Zhuonan Li, Joseph Hughes, Sam J. Wilson.

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
