## [Editor Report · Decision Letter 0]

31 Mar 2023

Dear Dr. Lytras, 

Thank you for submitting your manuscript entitled "Resurrecting an ancient horseshoe bat defence against SARS-related coronaviruses" for consideration as a Short Report by PLOS Biology.

Your manuscript has now been evaluated by the PLOS Biology editorial staff, as well as by an academic editor with relevant expertise, and I am writing to let you know that we would like to send your submission out for external peer review.

Once your full submission is complete, your paper will undergo a series of checks in preparation for peer review. After your manuscript has passed the checks it will be sent out for review. To provide the metadata for your submission, please Login to Editorial Manager (https://www.editorialmanager.com/pbiology) within two working days, i.e. by Apr 02 2023 11:59PM.

Kind regards,

Paula

---

Senior Editor

PLOS Biology

---

## [Decision Letter · Decision Letter 1]

5 May 2023

Dear Dr. Lytras,

Thank you for your patience while your manuscript "Resurrecting an ancient horseshoe bat defence against SARS-related coronaviruses" was peer-reviewed at PLOS Biology. It has now been evaluated by the PLOS Biology editors, an Academic Editor with relevant expertise, and by several independent reviewers. 

In light of the reviews, which you will find at the end of this email, we would like to invite you to revise the work to thoroughly address the reviewers' reports.

As you will see below, reviewer #3 wants you to demonstrate experimentally which mutations, other than the prenylation motif, are involved in the inactivation of antiviral activity against SARS-CoV-2. We think that you should address all the concerns from reviewer #3.

Given the extent of revision needed, we cannot make a decision about publication until we have seen the revised manuscript and your response to the reviewers' comments. Your revised manuscript is likely to be sent for further evaluation by all or a subset of the reviewers.

**IMPORTANT - SUBMITTING YOUR REVISION**

*Re-submission Checklist*

*Published Peer Review*

*PLOS Data Policy*

*Blot and Gel Data Policy*

Sincerely,

Paula

---

Senior Editor

PLOS Biology

REVIEWS:

Reviewer #1: Human pathogens and cellular factors interactions.

Reviewer #2: Virus-cell factors interactions.

Reviewer #3: Virus-cell factors interactions.

Reviewer #1: In this manuscript, Lytras and collaborators reveal that the loss of prenylation in horseshoe bat OAS1 causes this protein to lose its antiviral activity. However, OAS1 from a predicted common ancestor of Rhinolophoidea bats would retain prenylation and thus its antiviral activities. The authors also identified residues in Rhinolophus bat OAS1 under diversifying selection, which may be responsible for acquiring an alternative function in horseshoe bats, since the addition of prenylation signal did not rescue the antiviral activity of this protein. This is an exciting finding that is well supported by the data presented by the authors, and may help explain why rhinolophus bats are a major reservoir of coronaviruses. This reviewer has no concerns and recommends the manuscript for publication

Reviewer #2: Oligoadenylate synthetase (OAS) proteins can sense viral double-stranded RNA (dsRNA), which activates RNase L by catalyzing the production of 2′-5′-oligoadenylate, resulting in cleavage of viral RNA and inhibition of viral infection. OAS1 has been identified as an interferon-stimulated gene (ISG) that strongly inhibits SARS-CoV-2 infection. In addition, this anti-SARS-CoV-2 activity requires prenylation of OAS1, which is determined by a CAAX box prenylation motif at the C-terminus of this host protein.

In this manuscript, Lytras et al. used bioinformatic approach to analyze OAS1 proteins in horseshow bats, where SARS-CoV-2 might be originated from. They found that OAS1 proteins lost the prenylation signal and did not exhibit any anti-SARS-CoV-2 activity. When the prenylation signal was repaired, these OAS1 proteins re-gained this antiviral activity. Although these findings are interesting, they simply re-confirm the important role of prenylation in OAS1 antiviral activity. In addition, although these results may explain how SARS-CoV-2 could evolve from the Rhinolophoidea, the significance of these findings is not sufficient to attract the broad audience of this journal. 

Reviewer #3: General

The authors previously reported that Rhinolophus bats' OAS1 lost its antiviral activity against SARS-CoV-2 due to the truncation of the C-terminal region, resulting in the loss of prenylation motif. In this study, the authors reconstructed the ancestral sequence of Rhinolophus bats' OAS1 and revealed that the OAS1 of Rhinolophus bats' ancestors had antiviral activity against SARS-CoV-2. Furthermore, they showed that not only the loss of prenylation motif but also the various mutations accumulated during the evolution of Rhinolophus bats could be involved in the inactivation of antiviral activity against SARS-CoV-2. However, they have not experimentally demonstrated which mutations, other than the prenylation motif, are involved in the inactivation of antiviral activity against SARS-CoV-2. The authors should address this point by experiments.

Major

Major Comment 1

The authors should clearly state the specific objective of this study in the last paragraph of the introduction. In the second-to-last paragraph of the introduction, the authors present a hypothesis regarding the evolutionary story of Rhinolophus bats' OAS1 and SARS-related CoVs. In the final paragraph, the authors state that they reconstructed the ancestral sequence of Rhinolophus bats' OAS1 and examined its properties in order to test this hypothesis. However, there seems to be a logical gap between the hypothesis and the strategy. The authors should define the specific objective and fill in the logical gap. In my understanding, the specific objective is to clarify how Rhinolophus bats' OAS1 lost its inhibitory activity against SARS-related CoVs. Is my understanding correct?

Major Comment 2

The authors point out the importance of the variable indel region in the inhibitory activity against SARS-related CoVs. Additionally, in reconstructing the RhinoCA sequence for experimentation, they insert the variable indel region from P. alecto into the sequence estimated through ancestral sequence reconstruction. However, the importance of the variable indel region in the inhibitory activity against SARS-related CoVs has not been experimentally verified. The authors should examine the effect of the deletion on the inhibitory activity against SARS-related CoVs using RhinoCA as backbone sequence.

Major Comment 3

In the discussion, the authors argue that the substitution in the 34th amino acid may be crucial for the inactivation of antiviral activity against SARS-related CoVs. This hypothesis should be experimentally tested. If the results from Major Comment 2 and 3 are obtained, the authors will be able to directly discuss how the ancestral OAS1 lost its antiviral activity against SARS-related CoVs through a series of events, which would increase the biological significance of this study.

Minor

Minor Comment 1

In the main text, the term "RhinoCA" is used in two different contexts: one to represent the sequence estimated through ancestral sequence reconstruction, and the other to represent the actual sequence used in the experiments (which is the sequence estimated through ancestral reconstruction with the addition of P. alecto's variable indel region and C-terminal end sequence). It would be better to use two distinct words to distinguish these two contexts.

Minor Comment 2

For Fig. 1A, the legend should be updated to indicate that the highlighted yellow section represents the CAAX box.

Minor Comment 3

In Fig. 3C, how about illustrating the 34th residue as well?

---

## [Editor Report · Decision Letter 2]

11 Oct 2023

Dear Dr. Lytras,

Thank you for your patience while we considered your revised manuscript "Resurrecting an ancient horseshoe bat defence against SARS-related coronaviruses" for publication as a Short Reports at PLOS Biology. This revised version of your manuscript has been evaluated by the PLOS Biology editors and the Academic Editor.

Based on our Academic Editor's assessment of your revision, we are likely to accept this manuscript for publication, provided you satisfactorily address the following data and other policy-related requests.

1. DATA POLICY:

A) Supplementary files (e.g., excel). Please ensure that all data files are uploaded as 'Supporting Information' and are invariably referred to (in the manuscript, figure legends, and the Description field when uploading your files) using the following format verbatim: S1 Data, S2 Data, etc. Multiple panels of a single or even several figures can be included as multiple sheets in one excel file that is saved using exactly the following convention: S1_Data.xlsx (using an underscore).

B) Deposition in a publicly available repository. Please also provide the accession code or a reviewer link so that we may view your data before publication. 

Regardless of the method selected, please ensure that you provide the individual numerical values that underlie the summary data displayed in the following figure panels as they are essential for readers to assess your analysis and to reproduce it: Figures 1B, 2ABC, 3CDE, and Supplementary Figure S1.

We require the original, uncropped and minimally adjusted images supporting all blot and gel results reported in an article's figures or Supporting Information files. We will require these files before a manuscript can be accepted so please prepare and upload them now. We will need this for Figures 2A, 3C.

3. Please provide a blurb which (if accepted) will be included in our weekly and monthly Electronic Table of Contents, sent out to readers of PLOS Biology, and may be used to promote your article in social media. The blurb should be about 30-40 words long and is subject to editorial changes. It should, without exaggeration, entice people to read your manuscript. It should not be redundant with the title and should not contain acronyms or abbreviations.

4. Please note that sole deposition of data or code to GitHub would not be compliant with our policies, as this could be changed after publication (https://journals.plos.org/plosbiology/s/data-availability). However, once the data is final, you can archive your publicly available GitHub data to Zenodo. Once you do this, it will also generate a DOI number that you can provide us with. See the process for doing this here: https://docs.github.com/en/repositories/archiving-a-github-repository/referencing-and-citingcontent.

5. We suggest a change in the title: "Resurrection of 2′-5′-oligoadenylate synthetase 1 (OAS1) from the ancestor of modern horseshoe bats blocks SARS-CoV-2 replication".

Please carefully read our guidelines for how to prepare and upload this data: https://journals.plos.org/plosbiology/s/figures#loc-blot-and-gel-reporting-requirements

We expect to receive your revised manuscript within two weeks. 

*Published Peer Review History*

*Press*

Sincerely,

Paula

---

Senior Editor,

pjaureguionieva@plos.org,

PLOS Biology

---

## [Editor Report · Decision Letter 3]

20 Oct 2023

Dear Dr. Lytras,

Thank you for the submission of your revised Short Reports "Resurrection of 2′-5′-oligoadenylate synthetase 1 (OAS1) from the ancestor of modern horseshoe bats blocks SARS-CoV-2 replication" for publication in PLOS Biology. On behalf of my colleagues and the Academic Editor, Frank Kirchhoff, I am pleased to say that we can in principle accept your manuscript for publication, provided you address any remaining formatting and reporting issues. These will be detailed in an email you should receive within 2-3 business days from our colleagues in the journal operations team; no action is required from you until then. Please note that we will not be able to formally accept your manuscript and schedule it for publication until you have completed any requested changes.

PRESS

Sincerely, 

Paula 

---

Senior Editor

PLOS Biology
